# Developing Christ as Consolatory Example in the Christ Encomium

**Alex W. Muir**

Faculty of Theology and Religion, University of Oxford, Oxford OX1 2JD, UK; alex.muir@theology.ox.ac.uk

**Abstract:** While Paul Holloway's scholarship on Philippians has been important, his classification of Philippians as a letter of consolation has gained relatively little traction. Interestingly, however, Holloway follows Karl Barth in labelling a large section of the letter, Phil 1:27–2:16, a 'hortatory digression', which could be seen to diminish the extent of consolation in this part of the letter. In this article, I seek to develop Holloway's work to argue that the Christ encomium in Phil 2:6–11 has elements of consolatory discourse that relates to other parts of the letter. Phil 2:6–11 illustrates and exemplifies how comfort (παράκλησις), consolation (παραμύθιον), and joy (χαρά) can be derived by individuals and communities in the face of opposition or destitution (cf. Phil 1:27–2:4). I propose that Christ undergoes a form of voluntary desolation in 2:6–8 but then receives something different from consolation in his glorious exaltation and the bestowal of the divine name. Although Paul and the Philippians will not receive universal worship like Christ, they can imitate him by following in this trajectory of becoming like God, thus receiving divine consolation and transformation.

**Keywords:** consolation; Philippians; joy

## 1. Reading Philippians as an Ancient Letter of Consolation

It is impossible to know for certain whether the apostle Paul, along with his associate Timothy, set out to compose a letter of consolation to the community in Philippi. We do know, however, from ancient rhetorical handbooks that such letters existed (Malherbe 1988, p. 33) and, on a more popular level, that shorter notes of condolence were exchanged between individuals in antiquity (Chapa 1998). The hypothesis that Philippians bears the hallmarks of a letter of consolation is a promising one. Paul, writing in restricted circumstances from prison,[1] offers perspectives on how grief (λυπή) can be replaced with joy (χαρά). The apostle repeatedly expresses his own joy in the Lord (Phil 1:18; 2:17; 4:10) and encourages the Philippians to rejoice (2:18; 3:1; 4:4). He also articulates the joy he feels towards the Philippians (1:4; 4:1) around the joy that he wants the Philippians to experience through progress in the faith (1:25) that will contribute to his own joy (2:2). Although Paul does not use the Stoic categories of passion (πάθος) and 'good-emotion' (εὐπάθεια),[2] it is well known that grief and joy, respectively, were key terms within these categories, which justifies seeing Paul as interacting in some way with broader ancient philosophical ideas.

The foremost advocate of Philippians constituting a letter of consolation is Paul Holloway, who, in his recent commentary (Holloway 2017, p. 2) and a growing number of articles, contends that Philippians is 'first and foremost a letter of consolation'. Yet, Holloway's proposal has gained relatively little traction. Other scholars who interpret Paul within an ancient philosophical context have come to the same conclusions as Holloway. Hans Dieter Betz views Philippians as a *praemeditatio mortis* with some consolatory aspects but contends that 'they are associated with more comprehensive issues' (Betz 2015, p. 133), and Troels Engberg-Pedersen focuses almost exclusively on παράκλησις as related to exhortation and moral progress (Engberg-Pedersen 2000). From a different direction that challenges situating Paul within an ancient philosophical and consolatory tradition, Ryan Schellenberg has argued that Paul's experience is closer to that of prisoners longing to see their community again than the internalised Stoic joy that a heroizing Christian tradition has seen in Paul since Acts (Schellenberg 2021).

While I dissent from some of Holloway's finer points of exegesis—notably that the Philippians became indolent out of grief for Paul's imprisonment—I agree that there are several places where Paul employs modes of consolation. One part of the letter where Holloway's proposal could be strengthened, however, is with reference to Phil 1:27–2:16. Holloway follows the theologian Karl Barth and others in seeing Phil 1:27–2:16 as a 'hortatory digression' (Holloway 2017, p. 83). Holloway argues that Paul's remark in 2:17 about his probable death—'if I am also poured on top of the sacrifice of your faithfulness (εἰ καὶ σπένδομαι ἐπὶ τῇ θυσίᾳ καὶ λειτουργίᾳ τῆς πίστεως ὑμῶν)'—represents the reality or the 'frank assessment' (Holloway 2017, p. 104) for which the Philippians are prepared in these verses via exhortation. While exhortation was an important later stage of the overall practice of consolation, looking past the earlier necessary stages of sympathy and comfort attenuates a consolatory reading.

In what follows, I assess whether we can identify more consolatory aspects in parts of Phil 1:27–2:16, especially the Christ encomium in Phil 2:6–11.[3] Holloway has joined other scholars in seeing this passage as paradigmatic (e.g., Hurtado 2004) and has highlighted how *exempla* were frequently deployed in ancient consolation literature (Holloway 2017, p. 115). If further notions of consolation can be found to unify this section and connect it to other sections of the letter, then I suggest that this will strengthen the thesis that the letter to the Philippians has consolation at its heart.

## 2. Phil 1:27–2:5: Consolatory Discourse before the Christ Encomium

Having narrated his own circumstances in Phil 1:12–26, the apostle turns back to his addressees. Paul's desire, even while 'absent (ἀπών)' (1:27), is that the Philippians become unified in spirit and soul in the face of 'those who oppose (τῶν ἀντικειμένων)' (1:28).[4] To reach this rhetorical goal, some degree of consolation is expedient so that the Philippians do not become 'frightened (πτυρόμενοι)' (1:28). The apostle advances three initial arguments that are intended to move the Philippians away from such fear.

Firstly, he offers an apocalyptic-oriented 'proof' (ἔνδειξις) that God will judge between the opponents and believers: the former will face destruction; the latter, including the Philippians, will face salvation (1:28).[5] Secondly, Paul prepares the Philippians for future trials: in Phil 1:29, suffering is described in terms of something gifted (ἐχαρίσθη) for the sake of Christ (cf. 2 Cor 12:9). It is to be expected that allegiance to Christ (τὸ εἰς αὐτὸν πιστεύειν) will entail suffering for Christ (τὸ ὑπὲρ αὐτοῦ πάσχειν). Thirdly, the Philippians face the same sort of contest or conflict (ἀγών) as Paul (1:30). Although their situation might not be as severe and ongoing as Paul's,[6] it is serious and comparable, and so the apostle sympathises with and comforts them.

While these three arguments combined provide evidence of consolatory rhetoric, such rhetoric is not expressed explicitly in conventional terms of consolatory discourse. In Phil 2:1–2, however, Paul employs highly distinctive consolatory discourse as he strengthens his appeal: 'if, therefore, there is any comfort in Christ (παράκλησις ἐν Χριστῷ), any consolation of love (παραμύθιον ἀγάπης), any fellowship of spirit, compassion and mercy, fill up my joy (πληρώσατέ μου τὴν χαρὰν) so that you may have the same mindset'. In other words, Paul directly associates conformity to Christ with comfort and consolation.[7] If this is apprehended, the community in Philippi will be united and filled with the spirit, which will, in turn, bring joy to an afflicted apostle.

Paul's resulting exhortation in Phil 2:2–4 then logically flows from divine consolation; he comforts and strengthens the Philippians by redirecting them to consolation in Christ. While there are parallels to Stoic paraenesis,[8] Paul's narrative is rooted in divine consolation rather than any sage-like qualities belonging to Paul. The question, then, is whether this consolatory discourse is picked up in the resulting Christ encomium: Christ is undeniably an example like Paul, Timothy, and Epaphroditus in this passage, but can he be viewed as a consolatory example?

### 3. Phil 2:6–8: Voluntary Desolation of Christ via Isaiah

The encomium is prefaced by an exhortation to the Philippians to 'have the same mindset (φρονεῖτε)' among themselves as in the example of Christ Jesus that follows on from the consolation that is found in him (2:5). I submit that the example of Christ extends the ethic of the preceding teaching: just as the Philippians ought to be 'considering (ἡγούμενοι) others as surpassing themselves' (2:3), Paul relates how Christ 'did not consider (ἡγήσατο)' his own mode of being equal to God as something to be held too tightly (2:6).[9]

Instead, Paul goes on to narrate how Christ divested himself of divine attributes and took on the appearance of a slave (2:7). Then, again in continuity with the 'humility' (ταπεινοφροσύνη) that had earlier been enjoined upon the Philippians (2:3), Paul relates how Jesus 'humbled himself (ἐταπείνωσεν ἑαυτόν)' (2:8). Through Christ's dramatic assumption of a human appearance[10] and his obedience in undergoing not only the logical end of the ἄνθρωπος, viz., death, but 'a death by crucifixion' (2:8), Paul represents what I call Christ's *voluntary desolation*.[11]

To substantiate this reading, we must establish whether there are any meaningful precedents for such an act of desolation. The operative term here, naturally, is 'humility'. As Eve-Marie Becker has successfully shown in her work on humility, although Paul introduces a neologism with the word ταπεινοφροσύνη, the wider ταπειν- word-group was taken up by a variety of writers in antiquity (Becker 2020, pp. 53–65). It is mostly in line with the writers of the LXX and Plato, however, that Paul endows this word-group with a positive ethical sense. Humility is something relational—as she puts it: 'a tool of interaction, which is to be conceptualised with reference to the *individual*' (Becker 2020, p. 59)—and consequently it is something that can be fomented and exemplified.

While I do not think that there are any particular allusions to the Hebrew Bible in this first part of the narrative, Deutero-Isaiah is noteworthy for its occasional ταπειν- language. At the start of Isaiah 40, Jerusalem famously receives consolation; indeed, according to the LXX, she is assured that 'her humiliation has been fulfilled (ἐπλήσθη ἡ ταπείνωσις αὐτῆς)' (Isa 40:2).[12] The only other instance of the noun ταπείνωσις in Deutero-Isaiah appears in the servant-song at Isa 53:8, where the prophet narrates how 'in humility, his judgement was taken away (ἐν τῇ ταπεινώσει ἡ κρίσις αὐτοῦ ἤρθη)'.[13] While the Philippians text does not explicitly mention judgement, the notion of Christ Jesus giving up his divine agency to relate to the condition of the ἄνθρωπος can be likened to the servant who was carried off to death for the people. In Deutero-Isaiah, however, this death is distinctly for Israel; there are two passages where the people of Israel (Isa 49:13) and Jerusalem (Isa 54:11) are humbled but receive a degree of divine consolation that is displayed elsewhere in the Hebrew Bible.[14]

This brief survey of Deutero-Isaiah highlights how consolation can naturally succeed humiliation or destitution. Moreover, in the case of the suffering servant, humility can represent an act of voluntary desolation that leads to death. While the Christ encomium in Philippians does not mention the concept of sin, let alone a transfer of sins, there are certainly parallels between what the suffering servant undergoes for Israel and what Christ undergoes for humanity. In short, even if it is not explicit, Jesus' divine self-transformation is presented as an act of voluntary desolation that aligns with some of Israel's salvation history as portrayed in Isaiah. What happens, however, when Paul *does* allude to Isaiah in the second part of the encomium narrative? Is voluntary desolation met with a degree of consolation?

### 4. Phil 2:9–11: Christ's Glorious Exaltation and Gift of the Divine Name via Isaiah

In his voluntary divestment of his divinity and act of desolation, Jesus descends further than any incarnated being because his starting point is higher. The messiah is returned, however, to this position in 2:9–11. Holloway himself recognises the different agents who effect these transformations: 'Christ initially effects his own transformation; however, his change back to a divine form is effected by God' (Holloway 2017, pp. 122–23). The ques-

tion that arises, however, is whether there is anything consolatory in the second part of the narrative.

While the account is deliberately laconic, Christ's voluntary desolation is only temporary: he dies physically on the cross, and then because of this, through a process of divine sublimation, 'God hyper-exalted (ὑπερύψωσεν) and gifted him (ἐχαρίσατο) the name above every name' (2:9). The language of χαρίζω evidently recalls Phil 1:29, where the Philippians were comforted that 'it was gifted' (ἐχαρίσθη) to them to suffer on behalf of Christ. This additional semantic link between 1:27–2:5 and 2:6–11 highlights how some degree of imitation is desirable and indeed possible,[15] but in 2:10–11, it is reinforced that Christ bears the divine name and receives universal worship and acknowledgement (2:10–11), which sets him above and apart from the Philippians.

In Phil 2:10–11, there is a more definite allusion to Deutero-Isaiah, specifically Isa 45:23, and the rewriting of a narrative about the exaltation of Jesus Christ. It is worth laying out the passages in tandem:

> Isa 45:23, LXX: οἱ λόγοι μου οὐκ ἀποστραφήσονται ὅτι ἐμοὶ κάμψει πᾶν γόνυ καὶ ἐξομολογήσεται πᾶσα γλῶσσα τῷ θεῷ

> My words will not be returned because every knee will bend to me and every tongue will confess to God.

> Phil 2:10–11: ἵνα ἐν τῷ ὀνόματι Ἰησοῦ πᾶν γόνυ κάμψῃ ἐπουρανίων καὶ ἐπιγείων καὶ καταχθονίων καὶ πᾶσα γλῶσσα ἐξομολογήσηται ὅτι κύριος Ἰησοῦς Χριστὸς εἰς δόξαν θεοῦ πατρός.

> So that in the name of Jesus, every knee might bend of those in heaven, on earth, and under the hearth, and every tongue might confess, 'Lord Jesus Christ', to the glory of God the father.

While many parts of Deutero-Isaiah contain underlying elements of grief that justify viewing them as consolatory passages, Isaiah 45 focuses on the glory of the God of Israel. Indeed, in Isa 45:25, the prophet remarks that 'from the Lord all the seed of the sons of Israel will be justified and in God glorified (ἐν τῷ θεῷ ἐνδοξασθήσονται)'. So, in Phil 2:10–11 too, the conclusion of the narrative centres 'on the glory of God (εἰς δόξαν θεοῦ)' alongside universal acknowledgement of Jesus Christ as Lord. Notions of joy (χαρά) are noticeably absent; instead, we encounter glory (δόξα).

This leads to the conclusion that there is less consolation in the second part of the Christ encomium. Yet, if, in Paul's narrative, Christ is the source of comfort and consolation (2:1), then it does not follow that he should be the recipient of consolation; Christ merits and receives something different, and that is glory.[16] It is Paul and the Philippians, however, who are the recipients and envoys of consolation. The apostle turns to the effects of Christ's example—including its consolatory elements—in the next section of the letter.

## 5. Phil 2:12–16; 3:20–21: Christ's Example for the Philippians: Transformation and Consolation

Becker notes how '[a]n ethos that is grounded in a success story and is developed with special rhetorical shaping spurs the reader to imitation (*imitatio*)' (Becker 2020, p. 70). This is precisely what takes place in Phil 2:12–16 when Paul moves from the exaltation of Christ to the ethical conduct of the Philippians in the absence of the apostle (and Christ). While the Philippians will not receive worship like Christ, who has been restored to the divine form by God, they are called to imitate Christ in their ethical conduct and, in doing so, become like him and conform to the divine image as children of God. As George Van Kooten notes, 'Christ appears among the believers, in the likeness of a visible human being, in order to render assimilation to him possible' (Van Kooten 2008, p. 212).[17]

Through Christ's example, the Philippians are to assume active responsibility for their salvation, but this is in collaboration with the agency of God (θεός … ὁ ἐνεργῶν), who provokes the appropriate action in the believer (2:13).[18] By imitating Christ, the Philippians can participate in his resurrected life both now and in the future. The manifestation of

this is transformed ethical conduct, including doing everything 'without grumbling and disputes' (2:14). The apostle goes on to elucidate how obediently following the example of Christ leads to present transformation (2:15):

> ἵνα γένησθε ἄμεμπτοι καὶ ἀκέραιοι, τέκνα θεοῦ ἄμωμα μέσον γενεᾶς σκολιᾶς καὶ διεστραμμένης, ἐν οἷς φαίνεσθε ὡς φωστῆρες ἐν κόσμῳ

> So that you might be blameless and pure, unblemished children of God amid a crooked and perverted generation, among whom you shine like luminaries in the cosmos.

The temporal emphasis in Phil 2:15 is unequivocally present. While some scholars see allusions here to Deut 32:5[19] and Dan 12:3,[20] Paul is describing the current commencement of a future transformation, which these texts do not perform in precisely the same way. Holloway surmises that 'in Phil 2:15 the promise is that Christ-believers can begin to experience angelification already in this life' (Holloway 2017, p. 134). Read this way, Paul exhorts the Philippians towards a future cosmological transformation,[21] which can be started in an imperfect, even oppressive, age and world. The gentile Philippians can accordingly also receive present comfort and consolation through receiving divine pneuma along with the seed of Abraham.[22]

In short, whether Paul is present or absent, the exemplary narrative about Christ that he supplies in Phil 2:6–11 is designed to provide sympathy for any destitution the Philippians experience, as well an exhortation to imitate and begin to become like Christ insofar as they are children of God. Yet, on these verses—and indeed all of Phil 1:27–2:16—I disagree with Holloway, who states that they primarily function as 'a request for consolation' (Holloway 2017, p. 102) on Paul's part. While there is some coordination of joy between Paul and the Philippians in Phil 2:17–18—'I rejoice and rejoice with you all; in the same way, rejoice and rejoice with me (χαίρω καὶ συγχαίρω πᾶσιν ὑμῖν· τὸ δὲ αὐτὸ καὶ ὑμεῖς χαίρετε καὶ συγχαίρετέ μοι)'—the apostle is more concerned with establishing a network of consolation with the exemplary Christ at the centre. Paul and the Philippians can subsequently bring consolation to one another, but in Phil 2:14–16, Paul portrays the mutual and conjoint consolation that can be derived through the transformation effected by Christ.

In the following sections of the letter, the apostle goes on to draw upon more *exempla* as part of his consolatory narrative: Timothy (2:19–24), Epaphroditus (2:25–30), and himself (3:4–14). These *exempla* culminate in a final vision where notions of transformation and consolation resurface in continuity[23] with the Christ encomium in Phil 3:20–21:

> ἡμῶν γὰρ τὸ πολίτευμα ἐν οὐρανοῖς ὑπάρχει, ἐξ οὗ καὶ σωτῆρα ἀπεκδεχόμεθα κύριον Ἰησοῦν Χριστόν, ὃς μετασχηματίσει τὸ σῶμα τῆς ταπεινώσεως ἡμῶν σύμμορφον τῷ σώματι τῆς δόξης αὐτοῦ κατὰ τὴν ἐνέργειαν τοῦ δύνασθαι αὐτὸν καὶ ὑποτάξαι αὐτῷ τὰ πάντα.

> Our citizenship belongs in the heavens, from where we are eagerly awaiting a saviour, Lord Jesus Christ, who will transfigure the body of our humiliation, making it conformable to the body of his glory, through the agency that also enables him to subject all things to himself.

In the face of sources of opposition in Philippi that aggrieve both Paul and the Philippians, Paul constructs a narrative of consolation that borrows language from earlier in the letter with relation to the exemplary Christ and casts him once more as the agent of transformation.[24] This time, however, there is a vision of some degree of glory for those who await the saviour, Lord Jesus Christ, and it is fitting that soon after these verses, discourse relating to joy (4:1, 4:4) resurfaces. Logically continuing where the encomium left off, the messiah emerges as an agent of transformation, which is consoling for Paul and the Philippians in their present circumstances.

### 6. Conclusions

Overall, in the foregoing discussion, I have contended that Phil 2:6–11 (the Christ encomium) portrays Jesus Christ as a consolatory example. This is in keeping with the consolatory discourse that has already appeared in the letter, particularly in Phil 2:1–2, where notions of comfort (παράκλησις), consolation (παραμύθιον), and joy (χαρά) all feature. The main reason that the messiah can be seen as a consolatory example is on account of his voluntary desolation. His act of self-transformation from divine to human form involves humiliation (ταπείνωσις). While there are neither assured allusions nor echoes to Deutero-Isaiah in Phil 2:6–8, it is instructive to see how Israel experiences humiliation but then receives a degree of consolation in her affliction and how the suffering servant in his humiliation has a divine prerogative, viz., judgement, taken away.

While the gifting of the divine name to Jesus recalls one of Paul's earlier consolatory arguments to the Philippians—namely, that it was gifted to them to suffer for Christ's sake (1:29)—there is admittedly less consolation in the second part of the narrative of the encomium. Since Christ is the source of consolation, there is no real need for him to receive consolation; instead, he is set above and apart from the Philippians as the Lord Jesus Christ and is glorified. In Phil 2:10–11, there is an allusion to the end of Isaiah 45, which emphasises the glory of the one God of Israel.

So, although Christ receives glory instead of consolation for his voluntary desolation, in the encomium, he provides a consolatory example for Paul and the Philippians. This is precisely what the apostle elucidates in Phil 2:12–16: amid opposition, the Philippians can become like God qua children of God through obedience and right ethical conduct and begin to be transformed into angelic and pneumatic form. Then, at the parousia of the Lord Jesus Christ, Paul narrates how he and the Philippians will undergo a total transformation from the body of humiliation to one that conforms to the glory of Christ (Phil 3:21). This consoling vision only makes sense in the light of the Christ encomium in which Christ, in human form, humbles himself, but is then exalted by God. Even if there are limits to the concept, I hope to have developed some arguments in favour of seeing consolation at the heart of the narrative and argument of Philippians. Yet, whether this makes it an entire letter of consolation is still hard to quantify.

**Funding:** This research received no external funding.

**Institutional Review Board Statement:** Not applicable.

**Informed Consent Statement:** Not applicable.

**Data Availability Statement:** No new data were created or analysed in this study. Data sharing is not applicable to this article.

**Acknowledgments:** I would particularly like to thank the editors of this journal, the anonymous reviewers, and members of the Oxford New Testament Research Seminar for feedback on this article.

**Conflicts of Interest:** The author declares no conflict of interest.

### Notes

[1]    I maintain that his imprisonment is in Rome, but I do not think that it affects my argument if it is in Ephesus, Caesarea, Corinth, or elsewhere.

[2]    On these categories, see Diogenes Laertius, *Lives* 7.116, and Cicero, *Tusc.* 4.6.13–14.

[3]    In this discussion, I remain agnostic about whether this is a hymn or not, but I shall refer to it as an encomium that was composed by Paul. In doing so, I draw upon (Basevi and Chapa 1993, p. 356): 'Phil. 2.5–11 should be viewed as an encomium of Christ which demands a poetical form and has the function of a profession of faith'. Holloway (Holloway 2017, p. 116) refers to '2:6–11 as a piece of elevated prose produced by Paul precisely for the exhortation of Phil 2:1–16'. While I consider the first part of this statement to be probable, I shall also argue that there is a more consolatory component in this section.

[4]    While my reading does not maximise unity as the purpose of the letter, it is an important feature, as Phil 4:2–3 confirms (see (Peterlin 1995)).

[5]    This sort of attitude is in line with broader Jewish apocalyptic literature, e.g., 4 Ezra 7.131 (NRSV): 'there shall not be grief at their destruction, so much as joy over those to whom salvation is assured'.

6    As (Von Gemünden 2015, p. 237) argues in her work on the affect of joy in Philippians: 'Trotz einer für die Gemeinde und noch deutlicher für Paulus schwierigen Situation'.

7    I view παράκλησις and παραμύθιον as related terms that belong to a similar semantic field. The latter term more consistently comes closer to 'consolation' as in, for example, 'the consolatory speech' (ὁ παραμυθητικὸς λόγος) known from the rhetorical handbooks (see Menander Rhetor, 413.3). The former term has a still-broader variety of meaning covering both consolation and exhortation (as well as more besides). I render it here as 'comfort' to bridge those terms: etymologically, comfort requires both a consoling presence and hortatory strengthening, which I consider to be the case in this context.

8    Thus (Engberg-Pedersen 2000, p. 217), 'the kind of community to which all of Paul's *paraklēsis* is directed … is nothing but an ideal community of friends, as the philosophers conceived of this'.

9    For a comprehensive treatment of the phrase τὸ εἶναι ἴσα θεῷ, see (Fletcher-Louis 2020).

10   Eastman (Eastman 2017, p. 130) comments how 'Christ "im-personates" Adamic humanity on the stage of human history'.

11   I see a similar exemplary move in the voluntary destitution represented by Christ's poverty as narrated by Paul in 2 Cor 8:9. On the deliberate compactness of this example, see (Mitchell 2017, p. 131); on the intended social effects of Christ's rich poverty, see (Barclay 2023).

12   Significantly, this differs from the MT: מָלְאָה צְבָאָהּ ('her warfare has been accomplished').

13   For further reference, see (Becker 2020, pp. 68–69).

14   (Bockmuehl 1997a, p. 21 n. 56) draws attention to a 'rich tradition of Jewish interpretation according to which God personally identifies with the suffering and affliction of his people', drawing on texts such as Exod 3:7f. and Isa 63:9.

15   Note how Epaphroditus in a later narrative in the letter imitates Christ: Just as Christ was obedient 'unto death (μέχρι θανάτου)' (2:8), Epaphroditus also 'approached unto death (μέχρι θανάτου)' (2:30). For further reference, see (Holloway 2017, p. 143).

16   This state of affairs is conceivably comparable to the Gospel of John: Jesus promises the distribution of χαρά to the disciples (Jn 15:11; 16:20–24; 17:13) but memorably is focused on the δόξα that he will receive from or with the father – notably in Jn 17:5. On the consolatory aspects of the Farewell Discourses, see (Parsenios 2005).

17   Like Van Kooten, I am sympathetic to the notion that Paul is participating in a discourse of 'becoming like God' that, although Platonic in origin, was taken up by other ancient philosophers and intellectuals. On the *topos* of 'becoming like God', see the excellent discussion by (Reydams-Schils 2017).

18   For further reference, see (Eastman 2017, p. 149): 'the divine agent has come near to energize them in the midst of their struggles'.

19   On issues with textual transmission, see (Bockmuehl 1997b, p. 156). Holloway (Holloway 2017, p. 134) is informative on how mapping Israel in Dt 32:5 onto this verse is inapposite and supersessionist.

20   The difference between φωστῆρες and ἀστέρες (Dan 12:3) is significant, and Daniel 12 speaks of transformation exclusively in the age to come.

21   Engberg-Pedersen (Engberg-Pedersen 2015, p. 303) provides a perceptive remark about the relationship between paraenesis and cosmology: 'The paraenesis (cognitive) *appeals to* the *pneuma* (both cognitive and material) that they already possess. And the aim is to bring about their final bodily transformation' (emphasis provided in the original).

22   For a comprehensive survey of how stars were viewed as divine in ancient Jewish and Graeco-Roman philosophical traditions, see (Thiessen 2016, pp. 140–47).

23   For a recent argument suggesting that Phil 3:20–21 is another liturgical fragment that originally followed Phil 2:6–11, see (Fletcher-Louis 2023, pp. 6–8).

24   For an analysis of how Phil 3:20–21 represents a moment of climactic consolation in the letter, see (Muir 2022).

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
