# Peer review of "Developing Christ as Consolatory Example in the Christ Encomium"

_religions, doi:10.3390/rel15050607_

Round 1
Reviewer 1 Report
Comments and Suggestions for Authors
On p. 1, the author implies the use of categories by Paul, i.e. pathos and eupatheia, as applied to the Greek terms lupe and khara in Philippians, which are interpretive categories of philosophers (Diogenes Laertius, etc.), not Paul's own terms.
On p. 3, the author translates terms from Phil 2:1, rendering paraklesis as 'comfort', next to rendering paramuthion as 'consolation'. This may be in the interest of the 'consolatory discourse' hypothesis, but paraklesis can also be rendered as 'encouragement' (e.g. Bockmuehl, RSV) and BDAG considers both possible for paraklesis in Phil 2:1. Additional argumentation for rendering 'comfort' would be in the interest of the overall argument. Otherwise one could also refer paraklesis to exhortation and hortatory literature.
Comments on the Quality of English LanguageIn the penultimate sentence, "I hope to developed" should be changed into "I hope to have developed".
Author Response
Many thanks for your interaction with my work.
In response to your comments:
1) I have tried to make it clearer that the categories of pathos and eupatheia are not directly used by Paul (see the highlighted section on p.1)
2) I have slightly conceded that paraklesis could mean something close to 'exhortation'. I have added a footnote (n. 7) which tries to parse a difference between comfort and consolation.

Reviewer 2 Report
Comments and Suggestions for Authors
Developing Christ as Consolatory Example in the Christ Encomium
Reviewer Report (Religions)
This article explores consolatory aspects in Philippians 1:27–2:16, especially with respect to the Christ encomium in 2:6–11. Building on Paul Holloway, the author seeks to demonstrate how the Christ encomium in 2:6–11 contains elements of consolatory discourse that may be related to other parts of the letter.
It might be helpful to mention three areas in which the article might be improved.
First, the author states, “By imitating Christ, the Philippians can assimilate to, and participate in, his resurrected life both now and in the future”(p. 6). It would be good to show how the Philippians participate in his resurrected life by imitating Christ. Also, the author seems to use the terms “assimilation” and “participation” synonymously without providing any specific definition of the theologically loaded terms.
Second, the author uses the language of becoming like God(p. 5). This language is often used in conjunction with the language of theosis in the patristic Christian tradition. If so, the author wants to nuance the difference between becoming like God and imitation, though they can be overlapped in some ways.
Third, the author only wants to cite Greek text when it advances his or her argument. If it is not so necessary for the argument, it would be good to reduce the use of Greek text in the writing.
Author Response
Many thanks for your interaction with my work.
I took your two main comments about assimilation/participating/becoming like God together. I agree that the assimilation language was unhelpful, so I have altered it in all cases apart from the original quotation from van Kooten. I have held onto the language of becoming like God particularly in view of tekna theou in Phil 2.15. I added an explanatory footnote in which I cite Reydams-Schils work on the subject and contend that Paul is participating in a similar discourse.
I have, in a few places, also cut down the amount of Greek I was using.

Round 2
Reviewer 2 Report
Comments and Suggestions for Authors
The author adequately addresses the issues I raised. I think the paper is now good for publication.